# Regulation of Epithelial Cell Functions by the Osmolality and Hydrostatic Pressure Gradients: A Possible Role of the Tight Junction as a Sensor

**DOI:** 10.3390/ijms20143513

**Published:** 2019-07-17

**Authors:** Shinsaku Tokuda, Alan S. L. Yu

**Affiliations:** 1Department of Respiratory Medicine, Graduate School of Medicine, Kyoto University, Kyoto 606-8507, Japan; 2Division of Nephrology and Hypertension, Department of Internal Medicine, University of Kansas Medical Center, Kansas City, KS 66160, USA

**Keywords:** tight junction, osmolality, hydrostatic pressure, cancer, sensor

## Abstract

Epithelia act as a barrier to the external environment. The extracellular environment constantly changes, and the epithelia are required to regulate their function in accordance with the changes in the environment. It has been reported that a difference of the environment between the apical and basal sides of epithelia such as osmolality and hydrostatic pressure affects various epithelial functions including transepithelial transport, cytoskeleton, and cell proliferation. In this paper, we review the regulation of epithelial functions by the gradients of osmolality and hydrostatic pressure. We also examine the significance of this regulation in pathological conditions especially focusing on the role of the hydrostatic pressure gradient in the pathogenesis of carcinomas. Furthermore, we discuss the mechanism by which epithelia sense the osmotic and hydrostatic pressure gradients and the possible role of the tight junction as a sensor of the extracellular environment to regulate epithelial functions.

## 1. Introduction

In multicellular organisms, epithelia act as a barrier to the external environment and contribute to maintain the homeostasis in the internal environment. The environmental conditions including osmolality and hydrostatic pressure (HP) constantly changes with the biological activity [1,2,3]. The epithelia are required to regulate their functions including transepithelial transport in accordance with the changes in the environmental condition to maintain the homeostasis in the internal environment. It has been reported that differences of the osmolality and HP between the apical and basal sides of the epithelia affects various epithelial functions as reviewed in this paper.

In this paper, we review these reports and illustrate the effects of osmotic and HP gradients in the regulation of epithelial functions. Furthermore, we examine the significance of the regulation in pathological conditions and discuss the mechanism by which epithelia sense the gradients of osmolality and HP.

## 2. Effects of Osmolality on the Epithelia and Endothelia

### 2.1. Regulation of Osmolality in the Body

When there is a concentration difference of a certain substance between a semipermeable membrane which is permeable to water but not to the substance, the substance exerts a driving force for the water movement and the driving force is called osmotic pressure or osmolality. Epithelia act as a barrier to the external environment and osmolality in the apical side of the epithelia constantly changes with the biological activity. For example, the osmolality in the apical side of an intestine dramatically and dynamically changes with diet and its digestion. The osmolality in the surface of an airway epithelium changes with secretion and moisture, and urine osmolality fluctuates in accordance with the regulation of osmolality in a body (see below).

In contrast, the osmolality inside the body is strictly regulated within a narrow range (275–295 mOsm/kg) [4,5]. The regulation of osmolality is mainly performed in the kidney except in aquatic animals. The excretion of free water in urine is regulated in the kidney, which is further regulated by hormones such as antidiuretic hormone (ADH) [6]. In the interstitium of the kidney, the osmotic gradient is formed from medulla toward cortex (high osmolality in medulla) [5,7]. The glomerular filtrate flows through renal tubules, and renal tubules go down to medulla and then make a sharp loop (Henle’s loop) and return to cortex. The permeability of water in renal tubules after the Henle’s loop is very low and the filtrate grows hypotonic as sodium chloride is reabsorbed. Then the renal tubules gather into collecting ducts and again go down to medulla [5,7]. In collecting ducts, water is absorbed in accordance with the amount of aquaporin (AQP) in the cell membranes and ADH regulates the amount of AQP in the apical cell membrane [6]. As such, the amount of free water excretion in urine is regulated and osmolality in the body is kept within a narrow range. The osmolality in urine ranges from 30 to 1200 mOsm/kg in humans [5,7].

Therefore, osmolality in the apical side constantly changes whereas osmolality in the basal side is kept in a narrow range in most epithelia. Then, what kind of effects does the osmotic change have on the epithelia?

### 2.2. Effect of Osmolality on Cell Volume and Other Cell Functions

The effects of osmolality on cells received attention as early as 1930s. In the 1960s, the regulation of cell volume after the osmotic changes in the extracellular environment was actively investigated by using erythrocytes. When the extracellular osmolality is lowered, water flows into the cells as most cell membranes of animal cells have high water permeability and results in cell swelling. Then, cells start to restore their volume to their original size by excreting potassium chloride (regulatory volume decrease: RVD). In contrast, when the extracellular osmolality is raised, water flows out from the cells and results in cell shrinkage. Then cells start to restore their volume by the uptake of sodium chloride (regulatory volume increase: RVI). The mechanism of osmotic cell volume regulation has been intensively investigated, and transporters and channels involved in RVD and RVI have been identified [8,9]. Furthermore, cell volume alteration by the osmotic changes have been known to have an impact on various cell functions including cell proliferation, cell apoptosis, metabolism, epithelial transport and migration [1,10].

### 2.3. Effects of the Osmotic Gradient Between Apical and Basal Sides on the Epithelia

As noted above, osmolality in the apical side changes constantly whereas osmolality in the basal side is kept in a narrow range. This explains why there are often differences of the osmolality between apical and basal sides in most epithelia. Interestingly, the difference of the osmolality (the osmotic gradient across the epithelia) is known to affect various epithelial functions in a different manner from osmotic volume changes. In this paper, we focus on the studies which clearly show that the osmotic gradient between apical and basal sides, but not increase or decrease of osmolality in both sides, affects epithelial functions.

#### 2.3.1. Jejunum

The osmolality in the apical side of the intestinal epithelium drastically changes with the diet, and the osmotic gradient affects transepithelial transport of a jejunal epithelium in a guinea pig [11]. Madara has reported that increase of the osmolality in the apical side with mannitol (up to 600 mOsm) elevates transepithelial electrical resistance (TER) whereas increase of the osmolality in the basal side has no effect on TER and increase of the osmolality in both sides slightly reduces TER [11]. Thus, the osmotic gradient from the apical to basal side is thought to elevate TER. TER reflects ion permeability across the epithelia. There are two pathways for ions across the epithelia: a transcellular pathway via apical and basal cell membranes and a paracellular pathway across intercellular space. The permeability of the paracellular pathway is regulated by tight junctions (TJs) [12,13,14]. To study the effect of the osmotic gradient on TJs, Madara investigated morphological changes in TJs by freeze-fracture electron microscopy. Apical hyperosmolality induces increase in number and depth of TJ strands (Figure 1A, [11]). Thus, the osmotic gradient from apical to basal side increases transepithelial ion permeability with the morphological changes in TJs in the jejunum. These changes may contribute the regulation of ion absorption in the jejunum.

#### 2.3.2. Skin

In the skin, osmolality in the apical side changes with the condition of sweat and drying, and the osmotic gradient affects TER in a frog skin [19,20]. The increase of osmolality in the apical side with mannitol, acetamide, or thiourea reduces TER. In contrast, increase of osmolality in the basal side elevates TER. Increase of osmolality in both sides has almost no effect on TER [19]. The decrease of TER with apical hyperosmolality and the increase of TER with basal hyperosmolality is also reported in another study in a frog skin [20]. Thus, the osmotic gradient from apical to basal side reduces TER whereas that from basal to apical side elevates TER in the skin.

#### 2.3.3. Retina

In the retina, a retinal pigment epithelium (RPE) separates retinal (apical) and choroidal (basal) environment and contributes to a blood–retinal barrier (BRB), which provides proper environment for photoreceptor cells. The osmolality in the choroidal side is higher than retinal side in the physiological condition. When the BRB is disrupted in the pathological conditions such as diabetic retinopathy, the osmolality in the retinal side increases and results in the accumulation of water and macular edema [21].

The osmotic gradient is reported to affect the electrophysiological property of the RPE [22]. To measure the electrophysiological property, a microelectrode was placed inside of and across the RPE and electrophysiological measurement including electroretinogram (ERG) was performed in chick retinas. Apical hyperosmolality with 25 mM mannitol induces depolarization of the basal cell membrane with decrease of membrane resistance and amplifies light-evoked c-wave in ERG. Basal hyperosmolality has opposite effects on these measurements and hyperosmolality in both sides has no effect on c-wave in ERG. Thus, the osmotic gradient affects electrophysiological property of the RPE, which may have a role in the regulation of BRB in the physiological and pathological conditions.

#### 2.3.4. Vascular Endothelium

Endothelia in the brain form the blood–brain barrier (BBB) and restrict the permeation of substances into the brain. Infusion of osmotic agents such as mannitol into the carotid artery is known to induce transient increase of TJ permeability in the endothelia in the BBB [23,24], which is considered as the method to improve the drug delivery into the brain for the treatment of brain tumors and other brain diseases [25,26,27]. Interestingly, the osmotic gradient causes increase of permeability in the endothelia in bovine major cerebral artery [28]. Apical hyperosmolality with 20% mannitol increases albumin permeability in the endothelia. Basal hyperosmolality also increases albumin permeability. In contrast, hyperosmolality in both sides has no effect on albumin permeability. Thus, the osmotic gradient in both directions increases the albumin permeability in the brain endothelia, and the same mechanism may be involved in the increase of BBB permeability by the infusion of mannitol into the carotid artery.

#### 2.3.5. Bladder

As noted above, the urine osmolality fluctuates in accordance with the amount of free water excretion in the urine, thus the osmolality in the apical side of bladder changes with this fluctuation. The osmotic gradient affects the transepithelial transport of bladder epithelium. In toad bladder, apical hyperosmolality with urea increases the permeability of sucrose and water. Basal hyperosmolality or hyperosmolality in both sides has almost no effect on the permeability of sucrose and reduces water permeability [29]. Interestingly, apical hyperosmolality induces bleb formation between TJ strands (Figure 1B, [15]). The blebs are also formed in basal hypoosmolality but not in basal hyperosmolality, hyperosmolality in both sides, apical hypoosmolality or hypoosmolality in both sides. Thus, the osmotic gradient from apical to basal side increases permeability of sucrose and water with structural changes in TJs, which is thought to contribute the regulation of transepithelial transport in the bladder.

#### 2.3.6. Kidney, Distal Tubule

As noted above, the osmolality of apical and basal sides in renal tubules varies in each segment of nephron, and the osmotic gradient is known to affect transepithelial transport in *Xenupus* A6 cells, a model of distal tubule cells [16,17]. In A6 cells, basal hypoosmolality reduces TER. The permeability of sodium (P_Na_) is selectively increased than that of chloride (P_Cl_) (Figure 1C). A counterbalance of the osmotic gradient by adding sucrose eliminates the reduction of TER. Apical hyperosmolality also reduces TER with the selective increase of P_Na_ [16]. The osmotic gradient from apical to basal side also affects the localization of the claudin, a family protein of an integral membrane protein in TJs. Claudins are major constituent of tight junction strands and thought to be a major determinant of TJ permeability [30,31,32]. In A6 cells, claudin-1 is mainly localized in the entire lateral cell membrane and shows no colocalization with occludin (other TJ protein mainly localized at TJs). The osmotic gradient from apical to basal side alters the localization of claudin-1 to the apical end of the lateral membrane and claudin-1 shows colocalization with occludin (Figure 1C) [17]. In contrast, osmotic gradient from basal to apical side reduces TER with an equal increase in P_Na_ and P_Cl_ [16,17]. The selective increase of P_Na_ by the osmotic gradient from apical to basal side is also observed in Madin–Darby canine kidney (MDCK) I cells, a model of distal tubule cells [18]. Thus, the osmotic gradient from apical to basal side reduce TER with selective increase of P_Na_ in distal tubule cell models, and the regulation of transepithelial transport by the osmotic gradient may have a role in the regulation of ion reabsorption in the distal nephron.

#### 2.3.7. Kidney, Proximal Tubule

Sodium chloride is filtered in the glomerulus and approximately 70% of sodium chloride in the glomerular filtrate is absorbed in proximal tubules. The proximal tubules have high water permeability and the osmotic gradient created by the reabsorption of sodium generates the driving force for water absorption. However, the filtrate in the apical side of the proximal tubules flows before the osmolality in the apical and basal sides become equal and thus the osmolality in the basal side is slightly higher than the apical side in proximal tubules [33,34,35]. In addition, fluctuation of plasma osmolality in the physiological conditions such as the changes in sodium chloride or blood glucose concentration in the plasma affects the osmolality of the filtrate in the proximal tubules. The osmotic gradient is known to affect the permeability of TJs in MDCK II cells, a model of proximal tubule cells [18]. TJs of MDCK II cells have high ion permeability with cation selectivity. Claudin-2 is expressed in MDCK II cells as well as proximal tubule cells in vivo, and claudin-2 is known to form highly conductive channels with cation selectivity in TJ strands and to be a major determinant of the permeability property of TJs in MDCK II cells [36,37,38]. Interestingly, apical hypoosmolality induces a reduction of cation selectivity in MDCK II cells (Figure 1D). A counterbalance of the osmotic gradient by adding sucrose or mannitol eliminates the reduction of cation selectivity. Basal hyperosmolality also reduces cation selectivity and basal hypoosmolality, apical hyperosmolality, or hyperosmolality in both sides does not affect cation selectivity. The expression level of claudin-2 is not changed two hours after the apical hypoosmolality. Thus, the osmotic gradient from basal to apical side is thought to cause the reduction of cation selectivity in MDCK II cells. In addition, the osmotic gradient from basal to apical side induces bleb formation between TJ strands with the changes in actin filaments. Furthermore, the osmotic gradient from basal to apical side alters the shape of cell–cell contact in MDCK II cells from zigzag to more straight shape (Figure 1D). Interestingly, the reduction of cation selectivity, the bleb formation and the alteration in the shape of cell–cell contact are not observed in claudin-2 knockout MDCK II cells, suggesting that claudin-2 mediates these changes. Since claudin-2 is expressed in proximal tubules in vivo, the regulation of charge selectivity in TJs by the osmotic gradient may have a role in the regulation of ion reabsorption in the proximal tubules in the physiological conditions.

#### 2.3.8. Summary

Effects of the osmotic gradient on epithelia and endothelia are summarized in Table 1. The osmotic gradient is involved in the regulation of various cell functions in various epithelia and endothelia. The cell functions regulated by the osmotic gradient include transepithelial transport (paracellular transport in many cases), polarization and resistance of cell membrane, actin filaments and shape of cell–cell contact. Interestingly, the direction of the osmotic gradient which induces cell responses and the regulated cell functions are different among the types of epithelia. Therefore, it is speculated that epithelia distinguish the direction of the osmotic gradient and each epithelium has a different mechanism to regulate cell functions, which is likely to reflect a difference of physiological roles in each epithelium.

## 3. Effects of HP on the Epithelia and Endothelia

As shown in Table 1, the osmotic gradient is involved in the regulation of various cell functions in various epithelia and endothelia. If the permeability of the substances used in the studies in Table 1 across the epithelial sheets is lower than water permeability, the concentration difference of the substances is assumed to act as a driving force for the water movement across the epithelia. To put in perspective the magnitude of this, the concentration difference of just 1 mM osmotically active solute (fasting blood glucose level fluctuates from 3.9 to 6.1 mM in the physiological condition) is estimated to be 26 cmH_2_O. Surprisingly, HP less than 26 cmH_2_O has been reported to have great impact on epithelial functions as we will review here.

### 3.1. HP in the Body

As in the case of osmolality, HP in the apical side constantly changes in accordance with biological activity. For example, the HP in the apical side of intestine drastically changes with diet. HP in a gallbladder and a bladder changes with the amount of storage (bile and urine, respectively). In the lung, respiration affects the pressure in the airway and the pressure in the apical side of alveoli periodically becomes negative in inspiration and positive in expiration. In contrast, HP in the basal side (interstitial fluid pressure: IFP) is regulated in a certain range. Most loose tissues have slightly negative IFP (−1 to −3 cmH_2_O) and the negative value is relatively high in the lung IFP (−10 cmH_2_O). In contrast, IFP in the encased tissues such as kidney shows slightly positive values (0 to 2 cmH_2_O) [7,40,41]. Therefore, it is thought that HP is higher in the apical side in most epithelia in the physiological condition and the HP gradient between apical and basal sides fluctuates with the biological activity.

### 3.2. Effects of HP on Epithelia

#### 3.2.1. Intestine

In the intestine, HP in the apical side dynamically and drastically changes with diet, and water transport in the intestine is affected by the HP gradient. In the isolated dog jejunum and ileum, water is absorbed from the apical to the basal side in normal conditions. The HP from the apical to the basal side up to 20 cmH_2_O has no effect on water transport. In contrast, HP from basal to apical side reduces the water absorption dependent on the pressure and the direction of water movement is reversed at 4 cmH_2_O [42]. Similar effects of the HP gradient on water transport is observed in a small intestine in a hamster and a colon in a rat [43,44]. Thus, the HP gradient from basal side suppresses water absorption in the intestinal epithelia.

#### 3.2.2. Trachea

Chronic inflammation in the trachea such as asthma is known to increase permeability of airway epithelia. On the other hand, chronic inflammation is also known to increase IFP (see Section 3.4), and the HP gradient from the basal side affects the transepithelial transport in the tracheal epithelia. HP from basal side increases transepithelial conductance (reciprocal of electrical resistance) and permeability of mannitol and albumin in dog tracheal epithelium, transepithelial conductance and permeability of 70 kDa and 2000 kDa dextran in bovine tracheal epithelium, and permeability of 70 kDa dextran in guinea pig tracheal epithelium [45,46,47]. HP from apical from basal side has no effect on transepithelial transport in dog and bovine tracheal epithelia whereas it induces small increase in the permeability of 70 kDa dextran in guinea pig tracheal epithelium. Thus, the HP gradient from basal side increases the permeability of tracheal epithelium, which may be involved in the pathogenesis of asthma and other airway diseases.

#### 3.2.3. Alveolus

The HP in alveoli periodically alters with respiration. In addition, the pressure is thought to be affected in various clinical conditions such as acute respiratory distress syndrome (ARDS), interstitial pneumonia, and mechanical ventilation. The HP gradient affects transcellular transport in alveolar epithelia. In *Xenopus* alveoli, sodium is absorbed and chloride and potassium are secreted via the epithelial sodium channel (ENaC), chloride, and potassium channels in the apical cell membrane, respectively. The HP from the apical side reduces short-circuit current (Isc) from apical to basal side, the reflection of net ion transport across the epithelia. The HP from the basal side also reduces Isc whereas the increase of the HP in both sides has no effect on Isc, suggesting both directions of the HP gradients reduce Isc. The blocker of ENaC or a chloride channel enhances the reduction of Isc induced by the HP from apical side whereas the potassium channel blocker decreases the reduction of Isc [48,49]. Thus, the HP gradient from apical side increases sodium and chloride secretion and reduces potassium secretion via the transcellular pathway in the alveolar epithelia.

#### 3.2.4. Kidney, Distal Tubule

In the renal tubules, HP is applied from apical side by the flow of filtrate, and the HP gradient is known to affect various cell functions of *Xenopus* A6 cells [50,51]. The HP from basal side increases transepithelial conductance dependent on the degree of pressure. The HP from basal side also alters the structure of actin filaments and claudin-1 localization at the lateral side, increases cell height and stimulates transcellular chloride secretion (Figure 2A). The HP from apical side or the increase of the HP in both sides does not induce these changes. Thus, the HP gradient from the basal side affects various cell functions including transepithelial conductance, actin structure, claudin-1 localization, cell height, and transcellular transport in A6 cells.

#### 3.2.5. Glomerulus

In the glomerulus, glomerular epithelial cells (podocytes) play an important role in glomerular filtration. The podocytes have TJs in fetal period, but they develop slit diaphragms, special structure for the glomerular filtration, during the development, and lose TJs [53]. When podocytes are isolated and cultured in vitro, they dedifferentiate and lose various characteristics within 24 h, and they form TJs again. The glomerular filtrate flows from basal to apical side in a podocyte cell sheet in vivo, and the HP gradient from basal side has been reported to affect various characteristics of the podocytes [52]. When podocyte cells from a rat cell line are cultured on a membrane filter and 1 cmH_2_O HP is applied from basal side by changing the height of culture medium, the podocyte cell sheet forms a large whirl-like configuration compared with the control (no HP gradient) condition. Furthermore, podocyte cells show more round shape and have wide intercellular space and deep intercellular indentations (Figure 2B). The HP from basal side also induces the reduction of TER and the loss of keratine-18 expression, one of the dedifferentiation markers in the isolated podocytes, in some cells. Thus, the HP gradient from a the basal side affects various characteristics in podocytes, and it is speculated that these changes may be related to the redifferentiation of podocytes.

#### 3.2.6. Bladder

In the bladder, the HP from an apical side fluctuates with the amount of urine storage. The HP gradient is known to affect transepithelial transport of the bladder epithelium [54,55,56]. The HP from an apical side increases Isc from apical to basal side whereas HP from a basal side decreases Isc in the bullfrog bladder [54]. The increase of Isc by the HP from apical side is also observed in the rabbit bladder, which is inhibited by ENaC, chloride and potassium channel blockers [55,56]. Thus, the HP gradient from an apical side affects sodium absorption and chloride and potassium secretion in the bladder epithelium, which is thought to contribute to the regulation of ion absorption and excretion in the bladder.

#### 3.2.7. Mammary Gland

In mammary glands, HP in the apical side fluctuates with milk production and breastfeeding and the permeability of TJs in mammary epithelia is known to be affected by the status of milk storage in vivo. The HP gradient has been reported to affect transepithelial transport in cultured mammary epithelial HC11 cells [57,58]. The HP from basal side decreases TER and changes the direction of Isc. The increase of HP in both sides has no effect on TER. Thus, the HP gradient from a basal side affects TER and transcellular transport, which may be involved in the regulation of transepithelial transport in milk production and breastfeeding.

#### 3.2.8. Summary

Effects of the HP gradient on epithelia are summarized in Table 2. There are several reports which show that HP from the apical or basal side affects transepithelial transport in gallbladder, hepatocyte, nasal epithelia, and proximal tubule [59,60,61,62]. However, these reports do not have clear data that shows that the gradient is a definitive cause of the changes due to the purpose of the studies and/or the difficulty in an experimental system, and we do not include these studies in the Table 2. Nevertheless, there is a possibility that the HP gradient is also involved in the regulation of cell functions in these epithelia.

As shown in Table 2, the HP gradient is involved in the regulation of various cell functions in various epithelia. The cell functions regulated by the HP gradient include transepithelial transport (transcellular and paracellular transport), cytoskeleton and cell shape. Surprisingly, only several to several tens cmH_2_O HP gradients induce various cell responses, which are less than the osmotic pressure induced by the 1 mM concentration difference of the nonionized osmotic substance as a driving force for the water movement (26 cmH_2_O). In addition, the HP gradient and the osmotic gradient which are assumed to generate the water movement of a same direction induce different cell responses in some cases (see the cases in the jejunum and the distal tubule in Table 1 and Table 2). Therefore, the osmotic gradient and the HP gradient are thought to affect various cell functions through different mechanisms (see Section 4).

Furthermore, the direction of the HP gradient which induces cell responses and the regulated cell functions are different among the types of epithelia. Thus, it is speculated that epithelia distinguish the direction of the HP gradient and each epithelium has a different mechanism to regulate cell functions, which is likely to reflect the difference of physiological roles in each epithelium.

### 3.3. Effects of HP on Endothelia

Relatively high HP is applied to the endothelia from the apical side by the blood flow. The pressure is especially high in the artery (approximately 100 cmH_2_O), which fluctuates with heartbeats. There are many reports which show that the HP from apical or the basal side affects various endothelial functions such as transendothelial transport, cell adhesion, cell shape, intracellular Ca^2+^ concentration, expression of cytokines and so on [64,65,66,67,68,69,70,71,72]. On the other hand, the increase of 50–270 cmH_2_O HP to the whole cell by using specialized pressure chamber also affects various endothelial functions including transendothelial electrical resistance, cytoskeletons, cell proliferation, and expression of cytokines and cell adhesion proteins [73,74,75,76,77]. Furthermore, smaller HP (2 to 27 cmH_2_O) is also shown to have effects on F-actin and cell proliferation [78,79,80,81], although there is a report which shows that 136 cmH_2_O HP has no effect on F-actin and cell proliferation in endothelia [82]. Thus, it is important to distinguish the effect of the increase of the HP on whole cell and the effect of the HP gradient on endothelia. However, there is no study which clearly shows that the HP gradient has an effect on endothelial functions. Thus, there is a possibility that HP gradient affects endothelial cell functions, but currently we do not have enough evidence.

### 3.4. HP in the Pathological Conditions

The HP in the basal side of epithelia (IFP) is known to change in pathological conditions. For example, IFP is markedly decreased in burn to −27 to −42 cmH_2_O [83]. Edema induced by raising vascular pressure increases IFP in a dog lung [84]. On the other hand, acute inflammation induces interstitial edema but the IFP is rather decreased transiently by the reduction of tension in the interstitium due to physical denaturation of collagen [85]. In contrast, in chronic inflammation the IFP is known to be increased; experimental inflammation by the ligation of first maxillary molar in rat gingiva and experimental pulpitis in cat dental pulp induce the increase of IFP [86,87]. Meanwhile, the IFP is also increased in cancer tissues. The increase of IFP was reported in xenograft tumor in animals as early as 1950 [88,89,90]. The IFP is also increased in most human tumors to 14–54.4 cmH_2_O [91,92,93,94,95,96]. The precise mechanism of the increase of IFP in tumors is still incompletely understood, but fibrosis, fluid accumulation and increase of cell density is thought to be involved in the mechanism [97,98]. In addition, the increase of IFP is known to be the factor associated with poor prognosis in uterine cervical cancer [99,100].

It is worth noting that IFP is increased in both chronic inflammation and carcinomas. The causal association between chronic inflammation and carcinomas is reported in almost all organs [101]. The mechanism of the association is generally thought to be due to immune responses induced by the chronic inflammation; reactive oxygen species from immune cells contribute to the occurrence of gene mutation and cytokines secreted by the immune cells promote cell proliferation, inhibition of apoptosis, angiogenesis, metastasis and epithelial mesenchymal transition (EMT) of the carcinomas [101,102]. On the other hand, since IFP is increased in both chronic inflammation and carcinomas, it is thought that the increased IFP may also contribute the promotion of carcinomas in chronic inflammation.

#### 3.4.1. Effects of HP on Tumor Cells

The increase of HP in the extracellular environment by using a specialized pressure chamber is known to affect the characteristics of cancer cells. In osteosarcoma, the increase of 27 to 68 cmH_2_O HP induces promotion or suppression of cell proliferation dependent on cell lines. In the cell lines which show the promotion of cell proliferation by the HP, the HP increases sensitivity to cisplatin, elevates the expression of tissue plasminogen activator (TPA) and vascular endothelial growth factor C (VEGF-C), and suppresses the expression of VEGF-A [103,104,105,106]. In lung cancer cells (CL1-5 cells and A549 cells), 27 cmH_2_O HP increases migration speed and cell volume with the elevation of the expression of various proteins including AQP1, Snail, vinculin and caveolin-1 [107,108]. In urothelial carcinoma cells, 102–1020 cmH_2_O HP promotes apoptosis with the elevation of the expression of Fas ligand, toll-like receptor 6 and connective tissue growth factor in the presence of mitomycin C [109,110]. In addition, extremely high HP more than 1 million cmH_2_O induces immunogenic cell death in tumor cells and is used for the method in cancer immunotherapy [111,112]. In contrast, small degree of the HP gradient from a basal side has been known to have carcinogenic effects on the epithelia.

#### 3.4.2. Effects of the HP Gradient on Carcinogenic Properties of Epithelia

When epithelia are cultured on permeable filters and the HP gradient is applied from a basal side by changing the height of culture medium, the HP induces epithelial stratification in some epithelial cell lines including MDCK I cells, Caco-2 cells, a model of colon cancer cells, and EpH4 cells, a model of mammary epithelial cells (Figure 3, [113]). The stratification is not observed when the HP is applied from apical side or the HP in both sides are decreased or increased, suggesting the stratification is caused by the HP gradient from basal side. In MDCK II cells, the HP gradient from a basal side does not induce the stratification, indicating the responsiveness of epithelial stratification to the HP gradient is different among the cell types. The stratification continues to develop in the presence of the HP gradient from a basal side, and the elimination of the HP gradient restores the epithelia to an almost single layer in 2 days after the elimination.

Interestingly, cavities with the characteristics of apical cell polarity are formed within the stratified epithelia. Microvilli are observed at the surface of the cavities, and TJs with the functional barrier assessed by the biotin tracer experiment are formed between the cells surrounding the cavities. It is worth noting that a similar abnormality in cell polarity is observed in the epithelia expressing oncogene and in vivo cancer tissue. When K-ras is expressed in MDCK cells and the cells are culture on permeable filters, epithelial stratification is induced and cavities with microvilli and TJs are observed within the stratification [114]. Similar stratification is reported in rat salivary gland cells (Pa-4 cells) when Raf-1 is expressed and cultured on permeable filter [115]. Furthermore, small cavities with microvilli and TJs are observed within pulmonary metastases of the mammary adenocarcinoma induced by the infection of mammary tumor virus in mice, and the permeation of lanthanum nitrate is blocked by these TJs [116]. Therefore, the epithelial stratification and abnormal cell polarity induced by the HP gradient from a basal side are thought to be common characteristics observed in oncogene expressing cells and in vivo carcinomas.

In addition, the HP gradient from a basal side affects cell proliferation and apoptosis. The proportion of S phase cells assessed by the BrdU assay is elevated by the HP gradient from a basal side, indicating the acceleration of a cell cycle. The HP gradient from a basal side also reduces the amount of dead cells in culture supernatant, and the elimination of the HP gradient induces the emergence of numerous TUNEL positive cells from 6 h after the elimination, suggesting the suppression of apoptosis by the HP gradient. In addition, the HP gradient from a basal side decreases TER and increases P_Na_ with the increase of claudin-2 expression level. Furthermore, the epithelial stratification is suppressed by the activation of protein kinase A (PKA) and promoted by the inhibition of PKA [113]. Thus, the HP gradient from basal side induces abnormal cell polarity, accelerates cell proliferation and suppresses cell apoptosis, resulting in the epithelial stratification via the PKA pathway.

In contrast to the carcinogenic effects of the HP gradient from a basal side, the outcome of the treatment of the bladder carcinoma by the HP suggests the possibility that the HP gradient from an apical side may have an inhibitory effect on carcinomas.

#### 3.4.3. Treatment of the Bladder Carcinoma by HP from an Apical Side

In 1972, Helmstein developed and introduced a method for the treatment of the bladder carcinoma using HP. In this method, the pressure inside the bladder is kept at the level of diastolic blood pressure for 5–6 h under general anesthesia. Surprisingly, in the studies including the patients with pathological stage T4, the reduction of the tumor has been observed in 70%–100% and complete response has been achieved in 25%–45% [117,118,119,120,121]. The cause of antitumor effect in the HP treatment was speculated to be the ischemia induced by the HP; tumor cells were thought to be more sensitive to the ischemia by the HP than normal cells due to the vulnerability of tumor vessels and changes in metabolism in tumor cells, and the HP from apical side induced cell death only in tumor cells.

The current standard therapy of non-muscle invasive bladder carcinomas is transurethral resection of visible bladder tumor (TURBT) and intravesical instillation of chemotherapy or BCG dependent on the risk factors of recurrence, and the HP treatment for the bladder carcinoma is not performed in current practical clinic. The mechanism of the HP treatment of the bladder carcinoma is still poorly understood, but since the HP gradient from basal side has carcinogenic effects on epithelia and the elimination of the HP gradient induces cell apoptosis [113], it is likely that the HP gradient from apical side in the HP treatment itself may have suppressive effect on tumor cells and contribute to tumor reduction.

#### 3.4.4. Intervention of IFP in Tumors

If the HP gradient from basal side promotes the growth of carcinomas and the elimination of the HP gradient contributes to the treatment of carcinomas, the intervention of the HP gradient in tumor tissues may be used as the treatment of carcinomas. In the HP treatment of bladder carcinomas, the HP from the apical side is applied under general anesthesia, but it is practically difficult to change the HP in the apical side in other organs. In contrast, various interventions are known to affect the IFP in tumor tissues. In xenograft tumors in mice, hamsters or rats, knockout of Neural/glial antigen 2 (NG2) proteoglycan, hyperthermia (43 °C), and the treatments of the antibody against VEGF receptor-2, ZD6126 (tubulin-binding agent) and inhibitor of platelet-derived growth factor (PDGF) receptor reduce IFP whereas the treatment of angiotensin II elevates IFP [122,123,124,125,126,127]. However, the mechanism of the increase of IFP in tumor tissues is still not fully understood, and it is difficult to constantly keep the IFP in normal level. Thus, it is currently not feasible to eliminate the HP gradient with the intervention except for the bladder carcinoma. On the other hand, the elucidation of the mechanism in the regulation of epithelial functions by the HP gradient is thought to provide a clue for the alternative method to develop the treatment of carcinomas. Then, how do the epithelia sense the HP gradient?

## 4. Theoretical Speculation of the Mechanism about How Epithelia Sense Osmotic and HP Gradients

In physiological conditions, there are various differences between the apical and basal environments of the epithelia including osmolality, HP, concentration of various substances, and electrical potential. Steep gradients of osmolality, HP, concentration of various substances, and electrical potential between apical and basal sides are formed at the sites that act as barriers to these differences, and the major barriers are the apical and basal cell membranes and the TJs. It is reasonable to speculate that the sensor(s) of these differences must be located at the site of these steep gradients (Figure 4A). Thus, next we discuss the possibility that either the cell membranes or the TJs act as the sensor of the gradients.

### 4.1. Osmolality

The osmotic gradient induces various changes in epithelia (Table 1). It is shown in the studies in Table 1 that either apical or basal hyperosmolality induces cell responses but hyperosmolality in both sides does not induce these responses except for the study in gallbladder. Further, it is also shown that either apical or basal hypoosmolality induces cell response but hypoosmolality in both sides does not induce these responses in the bladder, proximal tubule and distal tubule cells [15,16,17,18]. In addition, the cell responses induced by the osmotic gradient are somewhat different from the responses induced by cell volume changes in hyperosmolality or hypoosmolality [1,10]. Therefore, epithelial cells are thought to sense and respond to the transepithelial osmotic gradient (i.e., the osmotic difference between the two sides) and not merely the overall extracellular osmolality.

First, we will consider the possibility that the apical cell membrane acts as a sensor of apical hyperosmolality. In the apical hyperosmotic condition, water flows out through the apical cell membrane (Figure 4B). In the condition of hyperosmolality in both sides, water also flows out through the apical cell membrane (Figure 4B). If the apical cell membrane is a sensor of the apical hyperosmolality and the water flow through the apical cell membrane induces cell responses, same cell responses should be induced by the hyperosmolality in both sides. However, the cell responses induced by the apical hyperosmolality are not induced by the hyperosmolality in both sides. Thus it is unlikely that the outflow of water through the apical cell membrane by itself is the trigger of cell responses induced by the apical hyperosmolality. Similarly, in the apical hyperosmotic condition, water flows in through the basal cell membrane. In the condition of hypoosmolality in both sides, water also flows in through the basal cell membrane. If the basal cell membrane is a sensor of the apical hyperosmolality and the water flow through the basal cell membrane induces cell responses, same cell responses should be induced by the hypoosmolality in both sides. However, the cell responses induced by the apical hyperosmolality in the bladder and proximal and distal tubule cells are not induced by the hypoosmolality in both sides. Thus it is unlikely that the inflow of water through the basal cell membrane by itself is the trigger of cell responses induced by the apical hyperosmolality (Figure 4B). There remains a possibility that apical and basal membranes act together as a sensor. If this is the case, the possible mechanism is that the water flow from both cell membranes induces the changes in cell membranes, cell shape and/or cytoskeleton and serves as a trigger. However, it may be difficult to distinguish between these changes in the osmotic gradient and those in cell swelling and shrinkage and provide enough sensitivity as a sensor. Therefore, it is not very likely that cell membranes are the sensor of the osmotic gradient.

In contrast, the possibility that TJs act as a sensor of the osmotic gradient is quite plausible. The movement of water and/or osmotic substances through the TJs is assumed to be sensed by TJs. One of the possible mechanisms in this case is that the movement of water and substances through the TJ strands act as shear stress and the shear stress is sensed by the TJ strands. Furthermore, the water movement may cause the accumulation of water between TJ strands because of differences in the water permeability between diffusion across TJ strands and diffusion through the paracellular space between strands, and this water accumulation may serve as a trigger of cell responses (Figure 4C). The bleb formation between TJ strands observed in the presence of the osmotic gradient is thought to support this possibility (Figure 1B, [15,18]). In addition, since TJ permeability is regulated by the osmotic gradient in many cases, TJs may constitute a feedback system in which TJs sense the extracellular environment and regulate the functions of themselves.

### 4.2. HP

The HP gradient also induces various changes in epithelia (Table 2 and [113]). Some studies in Table 1 show that these changes are not induced by the increase or decrease of HP in both sides [51,52,113]. In addition, the atmospheric pressure fluctuates more than 1 cmH_2_O in a day and more than 5 cmH_2_O from day to day, and it is unlikely that the fluctuation of atmospheric pressure induces the changes observed in Table 2. Thus, as in the case of osmolality, epithelial cells are thought to distinguish the HP gradient and the changes of HP in both sides.

In the condition of the HP gradient from basal side, it is thought that basal and apical cell membranes are pushed to apical side with the water movement (Figure 4D). In contrast, basal cell membrane is pushed to apical side and apical cell membrane is pushed to basal side with the water movement in the condition that the HP is increased in both sides, and the cell membranes are pushed to opposite direction with the water movement in the condition that HP is decreased in both sides. Thus, if the cell membranes act as a sensor of the HP gradient, it is unlikely that the push of basal cell membrane to apical side with the water movement by itself or the push of apical cell membrane to apical side by with the water movement itself is the trigger of the responses induced by the HP gradient (Figure 4D). There remains a possibility that the push of apical cell membrane to apical side and basal cell membrane act together as a sensor. If this is the case, the possible mechanism is that as in the case of osmolality, the push of cell membranes induces the changes in cell membranes, cell shape and/or cytoskeleton and serves as a trigger. However, it may be difficult to distinguish between these changes in the HP gradient and those in the increase or decrease of HP in both sides and provide enough sensitivity as a sensor. Therefore, it is not very likely that cell membranes are the sensor of the HP gradient.

In contrast, it is simple to consider the possibility that TJs act as a sensor of the HP gradient. The push of TJ strands to apical or basal side and movement of water through the TJs is sensed by the TJs. One of the possible mechanisms in this case is that the pressure to the TJ strands changes the structure of the intracellular proteins in TJ complexes and the tension applied to the cytoskeleton (Figure 5). The structural changes of F-actin in the lateral side and the increase of cell height by the HP gradient from basal side in *Xenupus* A6 cells may reflect the effect of HP gradient on the tension in F-actin (Figure 2A and Figure 4E, [51]).

In summary, it is likely that TJs act as a sensor of osmotic and HP gradients, although it is difficult to rule out the possibility that apical and basal cell membranes act together as a sensor. Lastly, we further discuss the possible role of TJs as a sensor

### 4.3. Possible Role of TJs as a Sensor

The TJs are one mode of the junctional complexes located in the most apical part of the complexes [12]. The TJs regulate the permeability of the paracellular pathway [13,14]. Claudins are a family protein of integral membrane proteins (26 members in human) in TJs and a major constituent of TJ strands [30,128,129]. Most epithelia express multiple claudins and expression patterns of claudins are thought to determine the permeability of TJs in each epithelium [31,32]. Most claudins have PDZ binding motif in carboxy-terminal tail and bind to scaffolding proteins in TJs which have PDZ motif such as ZO family proteins, multiple PDZ domain protein (MPDZ) and Pals1-associated tight junction protein (PATJ). These scaffolding proteins further bind to transmembrane proteins including claudins and other proteins, scaffolding proteins, F-actin and many other proteins involved in the regulation of various cell functions including cytoskeleton, transcription, signal transduction, and vesicular trafficking. Thus, TJs is a complex comprised of diverse proteins which are involved in the regulation of various cell functions including cell differentiation, proliferation and apoptosis [130,131,132,133,134].

As discussed above, the sensor of the environmental gradients between apical and basal sides of the epithelia is required to distinguish the direction of the gradient and the difference of the stimulation (such as osmolality and HP) to regulate various cell functions dependent on the physiological requirement in each epithelium. Since claudins are large family proteins and expression patterns of claudins in TJs are different among epithelial and endothelial cell types, TJ strands are thought to be quite diverse among cell types and likely to have the ability to distinguish the direction and the difference of the stimulation dependent on cell types. Furthermore, diverse proteins in TJ complex are thought to be a suitable system to transduce the stimulation sensed by the TJ strands to regulate various cell functions (Figure 5). So far, there is little evidence that shows that TJs act as a sensor of the environmental gradient between apical and basal sides. However, since the changes in the shape of cell–cell contact and F-actin induced by the osmotic gradient from apical to basal side in MDCK II cells are not observed in claudin-2 knockout cells (Figure 1D, [18]), claudin-2 is thought to mediate these changes by the osmotic gradient and likely supports the possibility that TJs act as a sensor. In addition, it is known that the expression pattern of claudins is changed in most carcinomas and changes in the expression level of some claudins affects cell functions such as cell proliferation [135,136]. Since the HP gradient from basal side has carcinogenic effects on epithelia, the altered expression pattern of claudins in carcinomas may contribute to transduce favorable signals induced by the HP gradient from basal side for cancer growth.

## 5. Conclusions

Osmotic and HP gradients have great impact on various cell functions in various epithelia. It is required to elucidate the mechanism about how epithelia sense the extracellular environment and regulate cell functions; the mechanism is thought to have physiological and cell biological significance and may contribute to the development of cancer therapy. Theoretical speculation suggests the possibility that TJs are involved in the sensing of osmotic and HP gradients, although obviously more evidence is required. Since TJs are intricate complexes composed of multiple proteins, it is speculated to be challenging to elucidate the mechanism due to the redundancy of many proteins and enormous protein interactions. We expect more examples that indicate the regulation of cell functions by the environmental gradient between apical and basal sides, which may provide a clue for the elucidation of the mechanism.

## Figures and Tables

**Figure 1 ijms-20-03513-f001:**
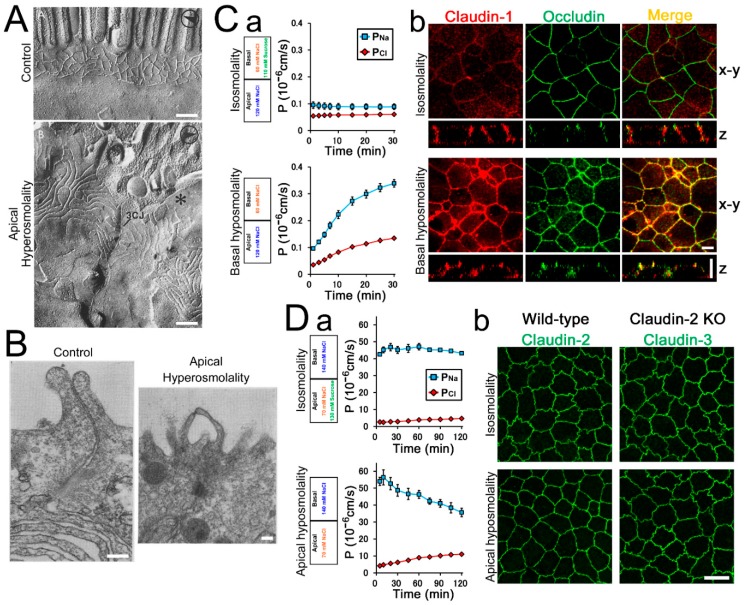
Effects of the osmotic gradient in various epithelia. (**A**) Freeze-fracture electron micrographs in the jejunal epithelium. Apical osmolality was increased to 600 mOsm with mannitol and cells were fixed 20 min after the osmotic changes. Apical hyperosmolality increased tight junction (TJ) strand number and depth. Scale bar = 200 nm. From Madara. *J. Cell Biol.* 1983 [11] with permission. (**B**) Transmission electron micrographs in the bladder epithelium. Apical osmolality was increased with 240 mM urea and cells were fixed 10 min after the osmotic changes. Apical hyperosmolality induced bleb formation between TJ strands. Scale bar = 200 nm. From Wade et al., *Am. J. Physiol.* 1973 [15] with permission. (**C**) Effects of basal hypoosmolality on *Xenopus* A6 cells. (**a**) Basal osmolality was decreased by the reduction of NaCl concentration or counterbalanced by the addition of sucrose, and permeability of sodium and chloride (P_Na_ and P_Cl_) were calculated from transepithelial resistance and dilution potentials in the presence of Na^+^, K^+^ and Cl^–^ channel blockers. Basal hypoosmolality increased P_Na_ and P_Cl_ with the selective increase of P_Na_. (**b**) Immunofluorescence of claudin-1 and occludin. Cells were fixed 30 min after the osmotic changes. Basal hypoosmolality altered claudin-1 localization to the apical end and claudin-1 showed colocalization with occludin. Scale bar = 5 µm. Modified from Tokuda et al., *Biochem. Biophys. Res. Commun.* 2008 [16] and Tokuda et al., *Biochem. Biophys. Res. Commun.* 2010 [17] with permission. (**D**) Effects of apical hypoosmolality on Madin–Darby canine kidney (MDCK) II cells. (**a**) Apical osmolality was decreased by the reduction of NaCl concentration or counterbalanced by the addition of sucrose. Apical hypoosmolality induced the reduction of cation selectivity. (**b**) Immunofluorescence of claudin-2 or claudin-3 in wild-type and claudin-2 knockout cells. Cells were fixed 30 min after the osmotic changes. Apical hypoosmolality altered the shape of cell–cell contact from zigzag to more straight shape in wild-type cells but not in claudin-2 knockout cells. Scale bar = 10 µm. Modified from Tokuda et al., *PLoS ONE.* 2016 [18] with permission.

**Figure 2 ijms-20-03513-f002:**
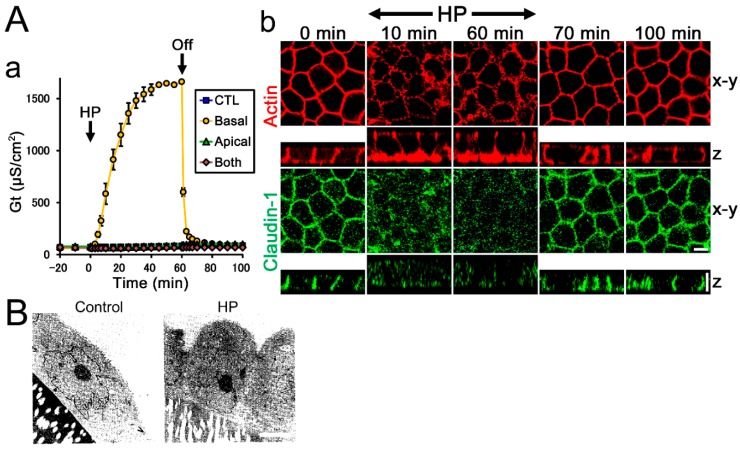
Effects of the hydrostatic pressure (HP) gradient in renal distal tubule cells and podocytes. (**A**) Effects of HP on *Xenopus* A6 cells. (**a**) Time course of transepithelial conductance. 8 cmH_2_O HP was applied from apical, basal, or both sides from time 0 to 60 min. The HP from basal side increased transepithelial conductance with reversibility. (**b**) Immunofluorescence of F-actin and claudin-1. 8 cmH_2_O HP from basal side was applied from time 0 to 60 min. HP from basal side increased cell height and altered actin structure and claudin-1 localization with reversibility. Modified from Tokuda et al., *Biochem. Biophys. Res. Commun.* 2009 [51] with permission. (**B**) Transmission electron micrographs of podocytes. 1 cmH_2_O HP was applied from basal side for three days. Podocyte cells showed more round shape and had wide intercellular space when the HP was applied from basal side. Scale bar = 5 µm. From Coers et al., *Pathobiology.* 1996 [52] with permission.

**Figure 3 ijms-20-03513-f003:**
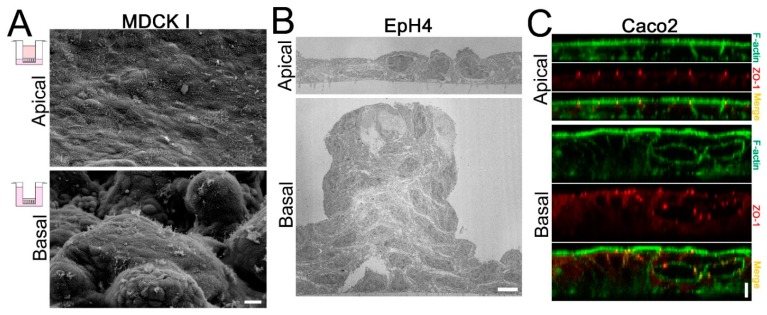
Effects of the HP gradient on carcinogenic properties of epithelia. (**A**) Scanning electron micrographs in MDCK I cells. 0.6 cmH_2_O HP was applied from apical or basal side for four days. A bumpy surface with cell masses was observed when the HP was applied from a basal side. Scale bar = 10 µm. (**B**) Transmission electron micrographs in EpH4 cells. 0.6 cmH_2_O HP was applied from apical or basal side for four days. HP from a basal side induced stratification. Cavities were observed within the stratification. Scale bar = 5 µm. (**C**) Immunofluorescence of F-actin and ZO-1 in Caco2 cells. 0.6 cmH_2_O HP was applied from apical or basal side for eight days. HP from a basal side induced stratification. ZO-1 was localized at the cavities within the stratification. Scale bar = 5 µm. Modified from Tokuda et al., *PLoS ONE.* 2015 [113] with permission.

**Figure 4 ijms-20-03513-f004:**
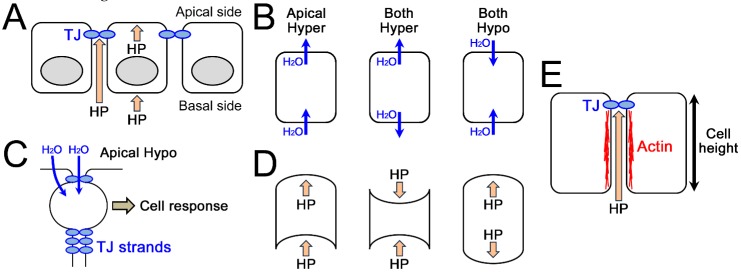
Theoretical speculation of the mechanism about how epithelia sense osmotic and HP gradients. (**A**) A model of epithelium in the HP from basal side. Steep HP gradients are formed at TJs and apical and basal cell membranes. (**B**) The water movement through apical and basal cell membranes in apical hyperosmolality (*left*), hyperosmolality in both sides (*middle*), and hypoosmolality in both sides (*right*). Hyper = hyperosmolality; Hypo = hypoosmolality. (**C**) Possible mechanism of bleb formation between TJ strands in apical hyperosmolality. (**D**) Effects of HP from basal side (*left*), increase of HP in both side (*middle*) and decrease of HP in both sides (*right*) on apical and basal cell membranes. (**E**) Possible effects of HP from basal side on F-actin in the lateral side and cell height.

**Figure 5 ijms-20-03513-f005:**
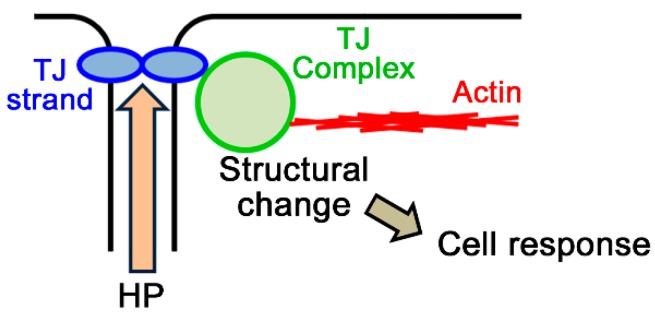
Possible mechanism of TJs as a sensor.

**Table 1 ijms-20-03513-t001:** Effects of the osmotic gradient on epithelia and endothelia.

Organ	Type of the Osmotic Gradient	Cell Response	References
Jejunum	Apical hyper (600 mOsm with mannitol)	TER, TJ strand number↑	[11]
Skin	Apical hyper (210 mM mannitol, acetamide, thiourea) Basal hyper (210 mM mannitol, acetamide, thiourea)	TER↓ TER↑	[19,20]
Retina (RPE)	Apical hyper (25 mM mannitol) Basal hyper (25 mM mannitol)	depolarization, Rt↓ in basal cell membrane polarization, Rt↑ in basal cell membrane	[22]
Brain (Endothelia)	Apical hyper (1100 mM mannitol) Basal hyper (1100 mM mannitol)	P_albumin_↑ P_albumin_↑	[28]
Bladder	Apical hyper (240 mM Urea) Apical hyper (240 mM Urea) or Basal hypo	P_sucrose_↑, P_H2O_↑ Bleb formation between TJ strands	[15,29]
Kidney (Distal tubule)	Basal hypo or Apical hyper (120 mM NaCl) Apical hypo or Basal hyper (120 mM NaCl)	TER↓, Claudin-1 localization to TJs TER↓	[16,17,18]
Kidney (Proximal tubule)	Apical hypo or Basal hyper (70 mM NaCl)	Cation selectivity↓, Bleb formation between TJ strands, Changes in cell–cell contact shape	[18]
Gallbladder	Apical hyper (100 mM sucrose)	TER↑, P_sucrose_↓, P_1,4-butanediol_↓	[39]

RPE = retinal pigment epithelium; Hyper = hyperosmolality; Hypo = hypoosmolality; TER = transepithelial electrical resistance; ↑ = increase; ↓ = decrease; TJ = tight junction; P_X_ = permeability of X; Rt = resistance.

**Table 2 ijms-20-03513-t002:** Effects of the HP gradient on epithelia.

Organ	Type of HP Gradient	Cell Response	References
Jejunum, ileum	Basal (20 cmH_2_O)	Water absorption↓	[42,43,44]
Trachea	Basal (5–20 cmH_2_O) Apical (20 cmH_2_O)	TER↓, P_mannitol_↑, P_water_↑, P_70kDa or 2000kDa dextran_↑, P_albumin_↑ P_70kDa dextran_↑	[45,46,47]
Alveolus	Apical (5 cmH_2_O) Basal (5 cmH_2_O)	Isc↓, K^+^ secretion↓, Na^+^ absorption↑, Cl^−^ secretion↑ Isc↓	[48,49]
Kidney (Distal tubule)	Basal (8 cmH_2_O)	TER↓, Cl^−^ secretion↑, Claudin-1 localization, Actin structure, Cell height	[50,51]
Kidney (Podocyte)	Basal (1 cmH_2_O)	Cell shape, TER↓, Expression of keratin-18↓	[52]
Bladder	Apical (1–8 cmH_2_O)	Na^+^ absorption↑, Cl^−^ secretion↑, K^+^ secretion↑	[54,55,56]
Mammary gland	Basal (10.2 cmH_2_O)	TER↓, Isc↓	[57,58]
Cervical epithelium	Basal (2.1 cmH_2_O)	TER↓, P_pyranine_↑	[63]

HP = hydrostatic pressure; TER = transepithelial electrical resistance; ↑ = increase; ↓ = decrease; P_X_ = permeability of X; Isc = short-circuit current.

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
