# Peer review of "Regulation of Epithelial Cell Functions by the Osmolality and Hydrostatic Pressure Gradients: A Possible Role of the Tight Junction as a Sensor"

_ijms, 2019, doi:10.3390/ijms20143513_

Round 1
Reviewer 1 Report
The review proposed by Shinsaku Tokuda et al is well written, and
comprehensively address the regulation of epithelial function by the
osmolality and hydrosatatic pressure gradient, in distinct epithelia.
The first part of the review describing either the osmolality or the hydrostatic pressure could be simplified to avoid the listing effect of the review.
The second part on the HP in pathological condition is
well describe and interesting.
The last part about the mechanisms of
how epithelia sense osmotic ans HP gradient is interesting but lack
significant prove of concept and the mechanism proposed is too
speculative.
Some re-organisation of the review has to be done to re-equilibrate each part.
Author Response
Reviewer 1
Point 1: The first part of the review describing either the osmolality or the hydrostatic pressure could be simplified to avoid the listing effect of the review.
Response 1: We thank for the helpful comment. We have removed the section about the effect of osmotic gradient on gallbladder (section 2.3.3) to simplify the osmolality section.
Point 2: The last part about the mechanisms of how epithelia sense osmotic ans HP gradient is interesting but lack significant prove of concept and the mechanism proposed is too speculative.
Response 2: We appreciate the reviewer’s very important advice. As we state in the responses 23, 25, 26, 27 and 28 in detail, we have mentioned that the osmotic change in the apical or basal side but not the osmotic change in both sides induces cell responses in the beginning of section 4. Further, we have illustrated the movement of water across the apical cell membrane in the apical hyperosmolality and hyperosmolality in both sides, and we have discussed why it is unlikely that the water movement through the apical cell membrane is not likely the trigger of cell responses induced by the osmotic gradient.
Point 3: Some re-organisation of the review has to be done to re-equilibrate each part.
Response 3: As we noted in the responses 1 and 2, we have simplified the section 2 and described in more detail the section 4 to re-equilibrate each part.
Reviewer 2 Report
Comments to the authors:
The manuscript by Tokuda and Yu titled “Regulation of epithelial cell function by the osmolality and hydrostatic pressure gradients: a possible role of the tight junction as a sensor” presents an overview of the effect of osmotic or hydrostatic pressure gradient on the structure and function of epithelial or endothelial cells. They provide a comprehensive review of the work being done to study these effects in various tissues as well as tumor cells. Most importantly, they compare the effect of changes in osmotic or hydrostatic pressure on basal and apical sides and conclude that the response of the cell to changes in pressure is not symmetric, i.e. the cell might respond to an increase in the basal pressure but not apical pressure.
The manuscript is concluded by speculating on possible mechanism of pressure sensing in the cell. The authors compare the possible response of cell membrane (through membrane transport) with that of tight junctions (paracellular transport) and conclude that tight junctions are more likely to act as the cell regulators for osmolality or hydrostatic pressure.
The manuscript is very valuable as it compares the effect of osmolality and hydrostatic pressure, and summarizes the current work in clear tables. The proposed mechanism of pressure sensing through tight junctions and its comparison with membrane transport is interesting. However, it is poorly written, specially the first segment that describes osmolality. Several paragraphs are missing important references or references are misplaced throughout the manuscript.
The last part on the sensing mechanism of tight junctions could also benefit from revisions (as detailed below) and could potentially be more concise and to the point.
Below is a list of my comments and suggestions that could improve the manuscript.
· The introduction could benefit from adding key references. Currently, the Introduction section does not include any citations. Including some of the key references on the general mechanisms that are discussed will be helpful. For example, it is mentioned that “differences of the osmolality and HP between the apical and basal sides of the epithelia affects various epithelial functions”. This is not a general fact and proper citation is needed. Or it is stated that “The external environment including osmolality and hydrostatic pressure (HP) constantly changes with the biological activity”. Some references here would be helpful, too. In addition, it would be more accurate to call osmolality or HP, environmental condition rather than external environment.
· On page 2, it is stated that “The excretion of free water in urine is regulated in the kidney, which is further regulated by hormones such as antidiuretic hormone (ADH).”
Later in the paragraph it is mentioned that: “In the interstitium of the kidney, the osmolality gradient is formed from medulla toward cortex.”
A few other very specific statements also describe the effect of osmolality in kidneys, …. However, strangely, this paragraph does not provide ANY references, even when specific values are given for osmolality in urine are mentioned. Proper citation for this paragraph needs to be provided.
· It is mentioned that osmolality in the body is strictly regulated within a narrow range (page 2). It will be helpful to provide a range or some numbers on how narrow this range is or how much the osmolality changes outside the cell to make the comparison more meaningful.
· Near the end of page 2, it is stated that “increase of the osmolality in the apical side with mannitol (up to 600 mOsm) elevates transepithelial electric resistance (TER) whereas increase of osmolality in the basal side has no effect on TER and increase of osmolality in both sides slightly reduces TER.” No reference is written for these statements. Do they refer to the jejunal epithelium that is referenced earlier? If so, the statement must be revised to reflect that or proper citation is needed.
· On page 3, it is stated that “the author investigated morphological changes in TJs by freeze-fracture electron microscopy”. Do they refer to their own work here? It is not clear which work they refer to, specifically since this manuscript has two authors (authors vs author) and immediately after they results of such work has been discussed. In addition, prior to this sentence another paper is cited, which makes it more confusing. Please clarify.
· On page 6, the authors start a section on “effects of HP on the epithelia and endothelia” by defining osmolality and comparing its effect with hydrostatic pressure. Even though it is nice to compare the two effects, this paragraph is poorly placed in the manuscript. The authors have dedicated the first six pages of the manuscript to review the effects of osmotic pressure in endothelia and epithelia, and on page six, they dedicate one complete paragraph to define it. I suggest moving this text to the beginning to section 2, before delving into the effect of osmolality in different organs and start this section only by comparing the magnitude of osmotic pressure and hydrostatic pressure.
Regarding the comparison, the authors compare the effect of 1mM nonionized substance to with 26 cmH2O (hydrostatic pressure). Since HPs less than 26 cmH2O have a great impact on epithelial function, they conclude that HP is important. A few things need to be clarified in this statement: (a) What is the significance of 1mM concentration of substance? It has be put into the context of substances in different organs or a comparison needs to be made, otherwise, this is not very meaningful, (b) when and where in the body HP less than 26 cmH2O is observed to have great impact? And most importantly proper citations for this claim is needed.
· The authors provide a review of the effect of osmotic and hydrostatic pressure in different organs. In particular, they describe the effect of these pressures on TER. It would be helpful if they could speculate what is the reason behind the increase or decrease in TER or other cell functions? Have any of the cited papers distinguish between e.g. increased expression of tight junction proteins versus changes in the properties of individual pores?
· On page 10, the authors state that “The HP in the basal side of epithelia (IFP) is known to change in pathological conditions. For example, IFP is markedly decreased in burn to -27 to -42 cmH2O.” Since such comparison are made throughout this paragraph for different tissues, it would be helpful to state what is the IFP in non-pathological conditions in each case.
· On page 11, the authors state that “The increase of HP to the whole cell by using a pressure chamber is known to affect the characteristics of cancer cells.” This statement is not accurate. What does it mean to increase the HP to the whole cell? Do the authors mean increasing the HP in the cell? Please rephrase?
· On page 4, section 2.3.3, the authors state that “Transepithelial transport of gallbladder epithelium is known to be affected by the osmotic gradient [15]” and continue with reviewing the results of experiments on sucrose permeability in rabbit gallbladder epithelium without any citation. It is not clear whether the authors are referring to the results presented in paper #15 or another paper. No other paper is cited in this paragraph. Since the starting statement is a very general statement not specific to any species, it is not clear what where the rabbit gallbladder results come from. Clarification is needed.
· On page 4, section 2.3.2, the authors discuss the effect of osmotic pressure in skin, referring to two papers (13, 14). However, the continue by reviewing very specific experiments showing these effects. It will help to cite the papers properly at the right place when talking e.g. about the increase of osmolality in the basal side on TER, … If both of these papers (13,14) provide the same results, it has to be cleared. In the current version, it is not clear whose work the authors are reviewing.
· On page 13, the authors state that “The gradients of these differences are formed at the sites which act as barriers of these differences, and the apical and basal cell membranes and tight junctions are thought to act as barriers of these difference.” This statement is not so clear, as they refer to differences (i.e. gradients) of differences between apical and basal environments. It needs to be revised to a more meaningful sentence.
In addition, later in the same paragraph they refer to the “boundary of the gradients”. As the gradient is defined over a region or between two regions, the boundary of the gradient is rather meaningless. Please revise.
· On page 14, the authors say that “most studies in table 1 show that these changes are not induced by hyperosmolality in both sides and some studies also show that these changes are not induced by hypoosmolality in both sides.” This statement needs clarification.
Do the authors mean that in most cases, the changes are induced by changing the osmotic pressure on one side, basal or apical, and not on both sides? And in cases that the pressure is changed on both sides, these effects are induced by increasing the pressure on one side and reducing it on the other side? This message does not come across from the current version of the manuscript, specially since the authors refer to these studies with “most studies” and “some studies”, there are only a handful of cases in table 1 that can easily be quantified.
Again, in the same paragraph, it is stated that “epithelial cells are thought to distinguish the osmotic gradient and osmotic gradient on both sides”. This statement is not clear, as the osmotic gradient is usually defined “between” two sides.
· On page 14, second paragraph, the argument provided by the authors is not very well written. The authors argue that osmotic gradient between apical side and inside of the cell, or basal solution and inside of the cell will result in water flow on each side. Then they argue that “thus, if the cell membrane acts as a sensor of the osmotic gradient, it is unlikely that the outflow of water through apical cell membrane by itself or the inflow of water through basal cell membrane by itself is the trigger of cell response induced by the osmotic gradient.” I could not follow the logic behind this statement. Further clarification is needed on why this flow cannot trigger the response of the cell.
The last argument on the same paragraph, that apical and basal cell membranes work together to change the shape of the cell or cytoskeleton. Then they argue that these changes can be the result of various conditions and are not exclusively the result of osmolality is more convincing and might be enough to prove their point.
· On page 14, the authors suggest that TJs can be considered as possible sensors of osmolality, by acting as flippers, without any further explanation. It is not clear what they mean by TJs acting as flippers. Clarification is needed.
The second part of their argument is also week. They argue that water transport through TJ strands might have different rates, and as a result, water might be accumulated between strands, and this accumulation of water might be sensed by the cell. Even though it is a nice hypothesis, it is hard to imagine why water permeability might be different between different TJ strands in the same cell. Another possible explanation for the bleb formation between TJ strands could be comparison of water permeability across TJ strands and diffusion rate of water in paracellular space.
· A similar argument is provided suggesting that hydrostatic pressure is also sensed at the TJ. I would like to add that HP can affect the water permeability similar to osmotic pressure. That is another possibility that needs to be discussed here and whether there is any evidence for that.
Author Response
Reviewer 2
Point 4: The proposed mechanism of pressure sensing through tight junctions and its comparison with membrane transport is interesting. However, it is poorly written, specially the first segment that describes osmolality. Several paragraphs are missing important references or references are misplaced throughout the manuscript.
Response 4: We thank for the helpful comment. We have added references and moved references to proper places as stated in each response to the reviewer comments below.
Point 5: The last part on the sensing mechanism of tight junctions could also benefit from revisions (as detailed below) and could potentially be more concise and to the point.
Response 5: We appreciate the extremely important advice. We have revised the manuscript as noted in the responses to the reviewer comments below.
Point 6: The introduction could benefit from adding key references. Currently, the Introduction section does not include any citations. Including some of the key references on the general mechanisms that are discussed will be helpful. For example, it is mentioned that “differences of the osmolality and HP between the apical and basal sides of the epithelia affects various epithelial functions”. This is not a general fact and proper citation is needed.
Response 6: We are sorry for the confusing description. We meant to mention one the most important conclusions in sections 2 and 3 here. We have added ‘‘as reviewed in this paper’’ at the end of the sentence.
Point 7: Or it is stated that “The external environment including osmolality and hydrostatic pressure (HP) constantly changes with the biological activity”. Some references here would be helpful, too.
Response 7: We thank for the helpful comment. We have added the references.
Point 8: In addition, it would be more accurate to call osmolality or HP, environmental condition rather than external environment.
Response 8: We appreciate for the marvelous advice. We have modified the expression of “external environment” to “environmental condition”.
Point 9: On page 2, it is stated that “The excretion of free water in urine is regulated in the kidney, which is further regulated by hormones such as antidiuretic hormone (ADH).” Later in the paragraph it is mentioned that: “In the interstitium of the kidney, the osmolality gradient is formed from medulla toward cortex.” A few other very specific statements also describe the effect of osmolality in kidneys, …. However, strangely, this paragraph does not provide ANY references, even when specific values are given for osmolality in urine are mentioned. Proper citation for this paragraph needs to be provided.
Response 9: We thank for the helpful comment. We have added the references.
Point 10: It is mentioned that osmolality in the body is strictly regulated within a narrow range (page 2). It will be helpful to provide a range or some numbers on how narrow this range is or how much the osmolality changes outside the cell to make the comparison more meaningful.
Response 10: We appreciate the helpful advice. We have added the normal range of plasma osmolality and a reference.
Point 11: Near the end of page 2, it is stated that “increase of the osmolality in the apical side with mannitol (up to 600 mOsm) elevates transepithelial electric resistance (TER) whereas increase of osmolality in the basal side has no effect on TER and increase of osmolality in both sides slightly reduces TER.” No reference is written for these statements. Do they refer to the jejunal epithelium that is referenced earlier? If so, the statement must be revised to reflect that or proper citation is needed.
Response 11: We are sorry for the inappropriate placement of the reference. These results are all from the reference [5]. We have added ‘[5]’ after this description.
Point 12: On page 3, it is stated that “the author investigated morphological changes in TJs by freeze-fracture electron microscopy”. Do they refer to their own work here? It is not clear which work they refer to, specifically since this manuscript has two authors (authors vs author) and immediately after they results of such work has been discussed. In addition, prior to this sentence another paper is cited, which makes it more confusing. Please clarify.
Response 12: We are sorry for the confusing description. “The author” we meant here was the author of reference [5]. To clarify the description, we have changed “the author” to the name of the author in [5] (Madara).
Point 13: On page 6, the authors start a section on “effects of HP on the epithelia and endothelia” by defining osmolality and comparing its effect with hydrostatic pressure. Even though it is nice to compare the two effects, this paragraph is poorly placed in the manuscript. The authors have dedicated the first six pages of the manuscript to review the effects of osmotic pressure in endothelia and epithelia, and on page six, they dedicate one complete paragraph to define it. I suggest moving this text to the beginning to section 2, before delving into the effect of osmolality in different organs and start this section only by comparing the magnitude of osmotic pressure and hydrostatic pressure.
Response 13: We thank for the helpful advice. We have moved the description about the definition of osmolality to the beginning of section 2.
Point 14: Regarding the comparison, the authors compare the effect of 1mM nonionized substance to with 26 cmH2O (hydrostatic pressure). Since HPs less than 26 cmH2O have a great impact on epithelial function, they conclude that HP is important. A few things need to be clarified in this statement: (a) What is the significance of 1mM concentration of substance? It has be put into the context of substances in different organs or a comparison needs to be made, otherwise, this is not very meaningful,
Response 14: We appreciate the important comment. There is fluctuation more than 1 mM of substances in the physiological condition. For example, fasting blood glucose fluctuates in the range from 3.9 to 6.1 mM and the plasma osmotic pressure fluctuates from 270 to 290 mOsm/kgH2O. We have added the fluctuation range of fasting blood glucose in the manuscript.
Point 15: (b) when and where in the body HP less than 26 cmH2O is observed to have great impact? And most importantly proper citations for this claim is needed.
Response 15: We are sorry for the confusing description again. We meant to mention about one the main conclusions in section 3 here. We have added ‘‘as we will review here’’ at the end of the sentence.
Point 16: The authors provide a review of the effect of osmotic and hydrostatic pressure in different organs. In particular, they describe the effect of these pressures on TER. It would be helpful if they could speculate what is the reason behind the increase or decrease in TER or other cell functions? Have any of the cited papers distinguish between e.g. increased expression of tight junction proteins versus changes in the properties of individual pores?
Response 16: We appreciate the very important comment. The changes in the expression and localization of tight junction proteins are investigated in a few papers. In MDCK II cells, apical hyposmolality reduces cation selectivity but the expression level of claudin-2 is not changed two hours after the stimulation. In MDCK I cells, HP from basal side induces the decrease of TER and the increase of Na+ permeability and the expression level of claudin-2 is increased. We have added these results in the manuscript.
Point 17: On page 10, the authors state that “The HP in the basal side of epithelia (IFP) is known to change in pathological conditions. For example, IFP is markedly decreased in burn to -27 to -42 cmH2O.” Since such comparison are made throughout this paragraph for different tissues, it would be helpful to state what is the IFP in non-pathological conditions in each case.
Response 17: We thank for the helpful comment. We have summarized the IFP in the physiological conditions in each tissue in the section 3.1.
Point 18: On page 11, the authors state that “The increase of HP to the whole cell by using a pressure chamber is known to affect the characteristics of cancer cells.” This statement is not accurate. What does it mean to increase the HP to the whole cell? Do the authors mean increasing the HP in the cell? Please rephrase?
Response 18: We thank for the helpful comment. We intended to mean the effect of the increase of HP in the extracellular environment by using specialized pressure chamber. We have revised the manuscript.
Point 19: On page 4, section 2.3.3, the authors state that “Transepithelial transport of gallbladder epithelium is known to be affected by the osmotic gradient [15]” and continue with reviewing the results of experiments on sucrose permeability in rabbit gallbladder epithelium without any citation. It is not clear whether the authors are referring to the results presented in paper #15 or another paper. No other paper is cited in this paragraph. Since the starting statement is a very general statement not specific to any species, it is not clear what where the rabbit gallbladder results come from. Clarification is needed.
Response 19: We thank for your helpful comment. The results of the effect of apical hyperosmolality in rabbit gallbladder are from [15]. However, the reviewer 1 required us to simplify the sections 2 and 3, and we have removed the section 2.3.3.
Point 20: On page 4, section 2.3.2, the authors discuss the effect of osmotic pressure in skin, referring to two papers (13, 14). However, the continue by reviewing very specific experiments showing these effects. It will help to cite the papers properly at the right place when talking e.g. about the increase of osmolality in the basal side on TER, … If both of these papers (13,14) provide the same results, it has to be cleared. In the current version, it is not clear whose work the authors are reviewing.
Response 20: We thank for the helpful comment. The results indicated here is from [13]. The decrease of TER by apical hyperosmolality and the increase of TER by basal hyperosmolality is also shown in [14] using the frog skin. We have cited [13] after the results and added the description about the results of [14].
Point 21: On page 13, the authors state that “The gradients of these differences are formed at the sites which act as barriers of these differences, and the apical and basal cell membranes and tight junctions are thought to act as barriers of these difference.” This statement is not so clear, as they refer to differences (i.e. gradients) of differences between apical and basal environments. It needs to be revised to a more meaningful sentence.
Response 21: We thank for the advice. We intended to mean ‘‘These differences’’ as the difference of HP, osmolality, various substances, potential difference, etc. between apical and basal environments. We have revised the manuscript.
Point 22: In addition, later in the same paragraph they refer to the “boundary of the gradients”. As the gradient is defined over a region or between two regions, the boundary of the gradient is rather meaningless. Please revise.
Response 22: We appreciate the critical advice. We have changed ‘‘the boundary of the gradient’’ to ‘‘the site of the steep gradients’’.
Point 23: On page 14, the authors say that “most studies in table 1 show that these changes are not induced by hyperosmolality in both sides and some studies also show that these changes are not induced by hypoosmolality in both sides.” This statement needs clarification. Do the authors mean that in most cases, the changes are induced by changing the osmotic pressure on one side, basal or apical, and not on both sides? And in cases that the pressure is changed on both sides, these effects are induced by increasing the pressure on one side and reducing it on the other side? This message does not come across from the current version of the manuscript, specially since the authors refer to these studies with “most studies” and “some studies”, there are only a handful of cases in table 1 that can easily be quantified.
Response 23: We appreciate your very importance advice. As the reviewer pointed, we meant that the osmotic change in the apical or basal side induces cell responses but the osmotic change in both sides does not induce these responses. It is shown that either apical or basal hyperosmolality induces the cell responses in the studies in Table 1 but hyperosmolality in both sides does not induce these responses except for the study in gallbladder. Further, it is also shown that either apical or basal hyposmolality induces cell responses but hyposmolality in both sides does not induce these responses in bladder, proximal tubule and distal tubule cells. We have added these in the manuscript.
Point 24: Again, in the same paragraph, it is stated that “epithelial cells are thought to distinguish the osmotic gradient and osmotic gradient on both sides”. This statement is not clear, as the osmotic gradient is usually defined “between” two sides.
Response 24: We are sorry for the confusing description. We have changed ‘‘osmotic gradient and osmotic gradient on both sides’’ to ‘‘osmotic gradient and osmotic changes in both sides’’.
Point 25: On page 14, second paragraph, the argument provided by the authors is not very well written. The authors argue that osmotic gradient between apical side and inside of the cell, or basal solution and inside of the cell will result in water flow on each side. Then they argue that “thus, if the cell membrane acts as a sensor of the osmotic gradient, it is unlikely that the outflow of water through apical cell membrane by itself or the inflow of water through basal cell membrane by itself is the trigger of cell response induced by the osmotic gradient.” I could not follow the logic behind this statement. Further clarification is needed on why this flow cannot trigger the response of the cell.
Response 25: We appreciate the extremely important comment and we apologize for the confusing description. We have modified the manuscript by illustrating the case of apical hyperosmolality. In the apical hyperosmotic condition, water flows out through the apical cell membrane. In the condition of hyperosmolality in both sides, water also flows out through the apical cell membrane. If the apical cell membrane is a sensor of the apical hyperosmolality and the water flows through the apical cell membrane induces various cell responses, same cell responses should be induced by the hyperosmolality in both sides. However, as we mentioned in the first paragraph in section 4.1, it is shown that apical hyperosmolality induces various cell responses but hyperosmolality in both sides does not induce these responses. Thus it is unlikely that the outflow of water through the apical cell membrane by itself is the trigger of cell responses induced by the osmotic gradient. We have explained the logic by illustrating the case of the effects of apical hyperosmolality on the apical and basal cell membranes, respectively, in the manuscript.
Point 26: On page 14, the authors suggest that TJs can be considered as possible sensors of osmolality, by acting as flippers, without any further explanation. It is not clear what they mean by TJs acting as flippers. Clarification is needed.
Response 26: We thank for the helpful comment. The movement of water or substances through the tight junctions is thought to act as shear stress, and we intended to describe the possibility that the shear stress is sensed by the tight junctions and mentioned it as ‘‘like flipper’’. However, as reviewer pointed, this description is rather confusing. We have revised the manuscript and mentioned the shear stress.
Point 27: The second part of their argument is also week. They argue that water transport through TJ strands might have different rates, and as a result, water might be accumulated between strands, and this accumulation of water might be sensed by the cell. Even though it is a nice hypothesis, it is hard to imagine why water permeability might be different between different TJ strands in the same cell. Another possible explanation for the bleb formation between TJ strands could be comparison of water permeability across TJ strands and diffusion rate of water in paracellular space.
Response 27: We appreciate the helpful suggestion. We agree with the reviewer comment. We have changed the description ‘‘if there is a difference in water permeability between TJ strands’’ to ‘‘because of differences in the water permeability between diffusion across TJ strands and diffusion through the paracellular space between strands’’.
Point 28: A similar argument is provided suggesting that hydrostatic pressure is also sensed at the TJ. I would like to add that HP can affect the water permeability similar to osmotic pressure. That is another possibility that needs to be discussed here and whether there is any evidence for that.
Response 28: We thank for the helpful comment. As the reviewer pointed, HP is also thought to induce water movement similar to osmolality. As mentioned in section 2.2, most cell membranes of animal cells have high water permeability and the change in HP is thought to induce water movement across the cell membrane. We have mentioned the water movement by the HP in the manuscript.
Reviewer 3 Report
The authors review the regulation of osmolality and hydrostatic pressure on epithelial cell functions and foresee the possible role of tight junction as sensor, opening an interesting topic to discuss. However, there are still several aspects to address before publication.
(1) There are many statements all across the manuscript without any citation, for example, introduction, and many other sections. This is a major problem for this review to be sound enough, and needs addressing one by one all across the body.
(2) The references are not latest enough, and the author need to add more new recent publications related with the topic.
(3) The figure legends need to rewrite in the style of brief conclusion instead of solely experimental description. This change can facilitate the reader's understanding of concepts.
(4) From page 3 to page 13, the author re-state the results of osmolality and HP's effect on epithelia of many organs, but what's the connection with the later section? What is the core information that the author want to pass on to the reader?
(5) On page 14, first three paragraph, the author's logic is hard to follow and very confusing. This need to clarify.
(6) Page 14, the last sentence, "it may be difficult to distinguish between..., therefore, it is not very likely that cell membrance are the sensor.." It is a fairly weak logic.
In conclusion, this review needs major revision to address above issues, and the authors need to pay special attention to update the references to the latest and include references for each individual statement.
Author Response
Reviewer 3
Point 29: (1) There are many statements all across the manuscript without any citation, for example, introduction, and many other sections. This is a major problem for this review to be sound enough, and needs addressing one by one all across the body. (2) The references are not latest enough, and the author need to add more new recent publications related with the topic.
Response 29: We thank the helpful comment. As we stated in detail in the response to the comments of reviewer 2, we have added latest reference in each section.
Point 30: (3) The figure legends need to rewrite in the style of brief conclusion instead of solely experimental description. This change can facilitate the reader's understanding of concepts.
Response 30: We appreciate your important comment. We have added brief conclusions in figure legends.
Point 31: (4) From page 3 to page 13, the author re-state the results of osmolality and HP's effect on epithelia of many organs, but what's the connection with the later section? What is the core information that the author want to pass on to the reader?
Response 31: We thank for the helpful comment. As we stated in detail in the responses 6, 15 and 23, one of the most important information in sections 2 and 3 is that the gradients of osmolality and HP but not the change of osmolality or HP in both sides induce cell responses in various tissues. This information is a critical basis for the discussion in section 4. We have further mentioned this information in the beginning of section 3 and added in the discussion in section 4.
Point 32: (5) On page 14, first three paragraph, the author's logic is hard to follow and very confusing. This need to clarify. (6) Page 14, the last sentence, "it may be difficult to distinguish between..., therefore, it is not very likely that cell membrance are the sensor.." It is a fairly weak logic.
Response 32: We appreciate the reviewer’s very important advice. As we state in the responses 23, 25, 26, 27 and 28 in detail, we have mentioned that the osmotic change in the apical or basal side but not the osmotic change in both sides induces cell responses in the beginning of section 4. Further, we have illustrated the movement of water across the apical cell membrane in the apical hyperosmolality and hyperosmolality in both sides, and we have discussed why it is unlikely that the water movement through the apical cell membrane is not likely the trigger of cell responses induced by the osmotic gradient.
Point 33: In conclusion, this review needs major revision to address above issues, and the authors need to pay special attention to update the references to the latest and include references for each individual statement.
Response 33: We thank for the helpful comment. We have added references as noted above.
Round 2
Reviewer 3 Report
I would agree to publish with the current edition.